# Mental health stigma in Ukraine over time: A cross-sectional study

Morgane Alexandra Petrie Gaschet[1] (iD), Orest Suvalo[1,2] and Vitalii Klymchuk[3] (iD)

[1]Mental Health for Ukraine, Ukraine; [2]Institute of Mental Health at the Ukrainian Catholic University, Lviv, Ukraine and [3]Department of Social Sciences, University of Luxembourg, Esch-sur-Alzette, Luxembourg

## Research Article

Mental health; Stigma; Ukraine

**Corresponding author:**
Morgane Alexandra Petrie Gaschet;
Email: Morganegaschet@outlook.com

## Abstract

This study examined changes in public knowledge, behaviours and attitudes towards individuals with mental health disorders in Ukraine. A nationwide survey was used to gather data from Ukrainian adults; this data was then compared with data gathered by Quirke et al. (2021, Cambridge Prisms Global Mental Health, 8) to form a comparison study. In congruence with the original study, the Mental Health Knowledge Schedule, the Community Attitudes towards Mental Illness Scale and the Reported Intended Behaviour scales were used. Measures of knowledge and attitudes towards individuals with mental disorders reflected a small reduction of knowledge (r = 0.13, p < .001) and a large reduction in benevolent attitudes (r = 0.96, p < .001). Conversely, there was a large decrease in authoritarian attitudes (r = −0.50, p < .001). Measures of behaviour reflected a medium positive increase in past and present behaviour (r = 0.33, p < .001) and a small positive increase in intended future behaviour towards individuals with mental illness (r = 0.24, p < .001). These findings provide a snapshot of changes in stigma towards those with mental health disorders in Ukraine and highlighted the growing need for evidence-based anti-stigma interventions and the monitoring of their impact.

## Impact statement

This study demonstrated varying changes in mental health stigma in Ukraine with regard to knowledge of, attitudes towards and behaviours towards those with mental health disorders. These have both improved and worsened since the escalation of the war and the delivery of numerous mental health anti-stigma campaigns in Ukraine. These findings hold implications for policymakers by highlighting the need to continue delivering campaigns aimed at targeting stigmatising views and behaviours towards those with mental health disorders. The study also highlights a strong need for monitoring of this type of stigma in Ukraine while campaigns are being delivered in order to better understand the impact of these.

## Introduction

The stigmatisation of individuals with mental health conditions is an evolving issue, which has been found across the world (Thornicroft et al., 2009). The recent Lancet Commission on ending stigma and discrimination in mental health posits that mental health stigma can be defined and understood across four different levels (Thornicroft et al., 2022). First, there is self-stigma; this refers to negative views individuals may have of themselves as a result of a mental health condition. Then, there is stigma by association; this refers to an individual with a mental health condition's internalisation of the negative views of others who are close with them. There is public stigma; this refers to the way in which a society or community views individuals with a mental health condition. And finally, there is structural stigma; this concerns the systems in place, such as laws, policies, cultures and organisations, and how they affect individuals with mental health conditions.

Efforts to reduce stigma are crucial to improve health and well-being outcomes for individuals with mental health conditions. A number of cross-sectional and longitudinal studies reflect that stigma is associated with both personal and clinical recovery outcomes, these include: feelings of hopelessness, overall reduced quality of life and other health-related outcomes (Livingston and Boyd, 2010; Schnyder et al., 2017; Dubreucq et al., 2020; Thornicroft et al., 2022). In young people, stigma is also associated with increased suicidality and self-harm (Thornicroft et al., 2022).

It has been noted that there is a gap in literature exploring mental health stigma in low- and middle-income countries (Morgan et al., 2018). In Ukraine, an upper middle-income country, a cross-sectional study of adults aged 18–60 found that while there was a degree of awareness, concern and compassion towards individuals with mental illness, there was also an overall high lack of knowledge and understanding of mental illness and treatments for these (Quirke et al.,

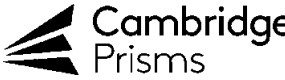



2021; Metreau et al., 2024). Moreover, a majority of respondents believed that community-based mental health services would downgrade neighbourhoods and present security risks. Overall, the study highlighted a strong need for anti-stigma interventions in Ukraine. However, since this study, circumstances in Ukraine have changed drastically.

These changes can be attributed to a changing landscape in the field of mental health, as well as the escalation of the war perpetrated by the Russian Federation. In 2020, Ukraine was found to have a similar prevalence of mental health disorders as other Eastern European countries (World Health Organisation, 2020). When compared globally, Ukraine was also reported to have a higher prevalence of alcohol use disorders and suicide rates. In addition to these findings, there is no doubt that the invasion of Ukraine by the Russian Federation, which escalated on the 24th of February 2022 has led to a mental health crisis (Seleznova et al., 2023). The Ukrainian population has been exposed to a range of traumatic events as well as displacement both within and outside of Ukraine, these factors have led to high rates of anxiety, depression, insomnia and post-traumatic stress disorder (Kang et al., 2023; Seleznova et al., 2023). Even prior to the escalation of the war, Ukraine had been identified as a country, which had limited resources with regard to meeting mental healthcare needs. Moreover, Ukraine was historically reliant on in-patient psychiatric services to provide mental healthcare rather than community-based mental healthcare, which they are now working towards (Skokauskas et al., 2020; Goto et al., 2023). However, the war has also greatly impacted mental health services in Ukraine. Some examples of the impact of the war include damage to the infrastructure of mental healthcare facilities, healthcare staff being physically injured or killed and staff members reporting an increase in mental health disorders and burnout at work (Goto et al., 2023). While it is clear that the war has impacted on the epidemiology of mental health and mental healthcare in Ukraine, it remains unclear whether there has also been an impact on mental health stigma.

In addition to changes brought on by the war, efforts to improve mental health knowledge and reduce stigma in Ukraine have been undertaken by a number of organisations in recent years. The Mental Health for Ukraine project has organised numerous information campaigns, which have involved the publication of posters and announcements in public spaces, social media posts, e-learning courses, educational articles, free videos and live seminars for adults and adolescents (Media Column, 2023; Mental Health for Ukraine, 2023; Center for Health and Development 'Family Circle,' 2024). Mental Health for Ukraine has also co-developed a campaign alongside the OneHealth consultant agency to support individuals who have been affected by sexual abuse (We Are Ukraine, 2024). Another key effort has been the development of the mental health ambassador network, which has supported individuals with lived experience to use their voices to educate others and influence policy development work (Mental Health for Ukraine, 2021, 2024). In addition to these, the First Lady of Ukraine, Olena Zelenska, has led a mental health awareness campaign known as "How are you?" which is coordinated by the Cabinet of Ministers of Ukraine Coordination Centre for Mental Health (The Presidential Office of Ukraine, 2024). The campaign was developed alongside key partners with the aim of improving public understanding and awareness of mental health.

While it is possible that all of the factors above have had an influence on mental health stigma experienced by individuals with conditions in Ukraine, no research has yet investigated whether this has been the case. The current study aims to investigate changes in mental health stigma in Ukraine through measures of knowledge of mental health, attitudes and behaviours towards individuals with disorders. Specifically, the study aimed to identify changes in: (1) public knowledge of mental health disorders and the treatment of these; (2) public attitudes towards individuals with mental health disorders and the treatment of these individuals within the community and (3) the public's intended behaviours towards individuals with mental health disorders. Moreover, in the same way that the original study measured these facets of stigma, the current study can act as a baseline for future monitoring of national trends in mental health stigma.

## Methods

### Design

Both the January 2020 and the June 2023 independent samples included in this paper were recruited by the same research organisation and therefore similar recruitment procedures were followed. The surveys were open to participants aged 18–60 who were based in both urban and rural regions of Ukraine. Participants were recruited using an online access panel of over 200,000 individuals and were interviewed using an online computer-assisted interview program. The pre-existing panel members from the access panel were all recruited through internet advertising as well as targeted in-person recruiting. Both of the current studies utilised a random selection of these existing panel members. As outlined in Ukrainian requirements, neither of the studies met the threshold for ethical approval; therefore, it was not required. However, all participants involved in the studies were required to provide consent prior to completing the surveys.

### Survey items

Both studies evaluated in this paper utilised the same measures as part of their surveys, these were the Mental Health Knowledge Schedule (MAKS), Community Attitudes Towards Mental Illness (CAMI) scale and the Reported and Intended Behaviour Scale (RIBS) (Taylor and Dear, 1981; Evans-Lacko et al., 2010; Evans-Lacko et al., 2011). Measures were initially translated into Ukrainian, reviewed by independent mental health experts, and back-translated into English for comparison by independent experts. Analysis of the internal consistency of the scales (Cronbach's α) was conducted during the first study in 2020 (Quirke et al., 2021).

#### Mental health knowledge schedule

The MAKS is a 12-item instrument that assesses stigma-related mental health knowledge (Evans-Lacko et al., 2010). The items are split into two subscales, with the first six items measuring health-seeking, recognition, support, employment, treatment and recovery. The other six items measure knowledge about specific mental health conditions. All items are answered using a five-point Likert scale (whereby a score of 1 = strongly disagrees with a proposed statement and 5 = strongly agrees). The total score range for each subscale is 6–30. For both subscales, a total score of each item is calculated, with a higher score indicating greater knowledge.

#### Community attitudes towards mental illness scale

The study utilised the CAMI scale to assess the attitudes, which people hold towards individuals with mental disorders living in the community (Taylor and Dear, 1981). The scale is made up of four

subscales, the first is Authoritarian, which encompasses questions related to the need to hospitalise the mentally ill, differences between those with and without mental illness, custodial care and the causes of mental disorders. The second subscale is Benevolence, these items focus on the need to be sympathetic and the responsibility of society to care for individuals with mental illness. The third is Social Restrictiveness, these items observe the degree of threat and danger, which individuals attribute to those with mental illness. And finally, the Community Mental Health Ideology subscale looks at attitudes towards the delivery of mental health services in the community.

Each of these subscales is made up of 10 items and can be responded to using a five-point Likert scale (whereby 1 = strongly agree and 5 = strongly disagree). The total score range for each subscale is 0–40. For the Authoritarian subscale, a total score is calculated, and a higher score indicates higher authoritarian attitudes towards individuals with mental illness. For the Benevolence subscale, a total score is calculated, and a higher score indicates higher benevolent attitudes towards individuals with mental illness. For the Social Restrictiveness subscale, a total score is calculated, and a higher score indicates higher restrictive attitudes towards individuals with mental illness. For the Community Mental Health Ideology subscale, a total score is calculated and a higher score indicates more positive attitudes towards delivering mental health services within the community.

The calculations of the CAMI scores for each subscale differ from the original ones used by Taylor and Dear (1981) to avoid the bipolar scales: "+20" was added to shift the scale from (−20… + 20) to scale (0…40). The same approach was utilised for the 2020 and 2023 data, with 2020 data being recalculated for this study.

### Reported and intended behaviour scale

The study utilised the RIBS to identify people's behaviour and intended behaviour towards individuals with mental illness (Evans-Lacko et al., 2011). The scale has two subscales, the first looks at past and present interactions, which people have with those with experience of mental illness, and the second looks at how individuals intend to interact with people with experience of mental illness.

The first subscale assessed past and present interactions with people with experience of mental disorders (4 items, yes/hard to say/no, coded from 0 to 2, respectively). The second subscale is comprised of four items, with the total score range being 4–20. The scale is answered using a five-point Likert scale (with 1 = strongly agree and 5 = strongly disagree). For both subscales, a total score of each item is calculated, with a higher score reflecting more positive behaviour towards individuals with mental illness.

### Data analysis

Data analysis was performed using JASP 0.14.3 (GNU Affero GPL v3 license, an open-source licence). Descriptive statistics (mean, standard deviation and frequency analysis) were used to describe the general results. Shapiro–Wilk test was used to check the normality of the data distribution. The Mann–Whitney $U$ test (independent samples, non-parametric as data were not distributed normally) was used to test statistical hypotheses about equivalences between independent samples. The effect size ($r$) was calculated by the matched rank biserial correlation due to the deviation of the data from the normal distribution.

## Results

### Sample demographics

In the first study (2020), a total of 1,007 individuals participated; in the second study (2023), a total of 1,050 individuals took part in the study. In both samples, the gender split of respondents was the same (51% female and 49% male). A more detailed breakdown of sample demographics can be seen in Table 1.

### Knowledge and awareness of mental health disorders

A Mann–Whitney $U$ test was conducted to determine if there were differences in mental health knowledge scores between the 2020 sample ($n = 1,007$) and the 2023 sample ($n = 1,050$). The test revealed a significant difference in scores between the two samples, $U = 598,212, Z = 5.99, p = <.001$. The effect size ($r$) was 0.13, this is a small effect size following Cohen's guideline (1988). The 2020 sample reflected a median score of 16 (95% CI [16.341, 16.708]), the 2023 sample reflected a median score of 16 (95% CI [15.728, 16.284]). These results suggest that public knowledge of mental health has reduced significantly between 2020 and 2023 (Table 2).

**Table 1.** Descriptive characteristics of the sample

|  | 2020 | | 2023 | |
| --- | --- | --- | --- | --- |
|  | N/mean | %/(SD) | N/mean | %/(SD) |
| Sample | 1,007 | 100% | 1,050 | 100% |
| Gender |  |  |  |  |
| Females | 514 | 51% | 537 | 51% |
| Males | 493 | 49% | 513 | 49% |
| Age | 39.2 | (10.95) | 40.0 | (11.22) |
| 18–30 | 262 | 26% | 252 | 24% |
| 31–40 | 292 | 29% | 294 | 28% |
| 41–50 | 222 | 22% | 273 | 26% |
| 51–60 | 231 | 23% | 231 | 22% |
| Region |  |  |  |  |
| North-Centre | 342 | 34% | 546 | 52% |
| West | 282 | 28% | 273 | 26% |
| East | 262 | 26% | 84 | 8% |
| South | 121 | 12% | 147 | 14% |
| Settlement type |  |  |  |  |
| City | 675 | 67% | 693 | 66% |
| Village | 332 | 33% | 357 | 34% |
| Education |  |  |  |  |
| Higher education | 547 | 54% | 515 | 49% |
| Other | 460 | 46% | 535 | 51% |

*Note*: North-Centre comprises Kyiv, Sumy, Vinnytsa, Zhytomyr, Kirovohrad, Chernihiv, Poltava and Cherkasy oblasts; West comprises Lviv, Ivano-Frankivsk, Khmelnytsky Rivne, Ternopil, Zakarpatie, Volyn and Chernivtsi oblasts; East comprises Dnipropetrovsk, Kharkiv, Donetsk and Zaporizhzhya and Luhansk oblasts; South comprises Odesa, Mykolaiv and Kherson oblasts. Higher education comprises those who have completed a bachelor, master or doctorate.

**Table 2.** Study result

| Measures | Median score (2020) | 95% CI (2020) | Median score (2023) | 95% CI (2023) | $U$ value | $Z$ value | $p$ value | Effect size ($r$) | Description |
|---|---|---|---|---|---|---|---|---|---|
| MAKS | 16 | [16.341, 16.708] | 16 | [15.728, 16.284] | 598,212 | 5.99 | <.001 | 0.13 | Small |
| RIBS: Past and present | 3 | [3.194, 3.377] | 2 | [2.127, 2.400] | 354,467 | −14.97 | <.001 | 0.33 | Medium |
| RIBS: Future intentions | 11 | [10.290, 10.779] | 12 | [12.071, 12.523] | 403,595 | −10.75 | <.001 | 0.24 | Small |
| CAMI: authoritarian | 18 | [18.153, 18.598] | 22 | [21.758, 22.246] | 266,486 | −22.49 | <.001 | −0.5 | Large |
| CAMI: Benevolence | 26 | [25.892, 26.412] | 13 | [12.878, 13.423] | 1,038,000 | 43.63 | <.001 | 0.96 | Large |
| CAMI: Social restrictiveness | 20 | [20.057, 20.615] | 20 | [20.034, 20.623] | 527,408 | −0.09 | 0.925 | 0.002 | Negligible |
| CAMI: Community ideology | 20 | [19.364, 19.951] | 19 | [18.508, 19.159] | 582,663 | 4.63 | <.001 | 0.1 | Small |

### Behaviours towards people with mental illness or in response to mental health disorders

#### Past and present behaviours

A Mann–Whitney $U$ test was conducted to determine if there was a significant difference in the past and present behaviour of individuals towards people with experience of mental health disorders between 2020 and 2023. The test revealed a significant difference in scores between the two samples, $U = 354{,}467$, $Z = −14.97$, $p = <.001$. The effect size ($r$) was .33, this is a medium effect size following Cohen's guideline (1988). The 2021 sample reflected a median score of 2 (95% CI [2.127, 2.400]), the 2020 sample reflected a median score of 3 (95% CI [3.194, 3.377]). These results suggest that reported past and present behaviours towards people with mental health disorders, which are positive have increased significantly between 2020 and 2023.

#### Intended future behaviour

A Mann–Whitney $U$ test was conducted to determine if there was a significant difference in the future intended behaviour of individuals towards people with experience of mental disorders between 2020 and 2023. The test revealed a significant difference in scores between the two samples, $U = 403{,}595$, $Z = −10.75$, $p = <.001$. The effect size ($r$) was .24, this is a small effect size following Cohen's guideline (1988). The 2020 sample showed a median score of 11 (95% CI [10.290, 10.779]) and the 2023 sample showed a median score of 12 (95% CI [12.071, 12.523]). These results suggest that intended future behaviours towards people with mental health disorders, which are positive have increased significantly between 2020 and 2023.

### Attitudes towards people with mental health disorders

#### Authoritarian

A Mann–Whitney $U$ test was conducted to determine if there was a significant difference in the authoritarian attitudes expressed towards individuals with mental illness between 2020 and 2023. The test revealed a significant difference in scores between the two samples, $U = 266{,}486$, $Z = −22.49$, $p = <.001$. The effect size ($r$) was −.50, this is a large effect size following Cohen's guideline (1988). The 2020 sample resulted in a median score of 18 (95% CI [18.153, 18.598]) and the 2023 sample resulted in a median score of 22 (95% CI [21.758, 22.246]). These results suggest that attitudes expressed towards individuals with mental illness, which are authoritarian have increased significantly between 2020 and 2023.

#### Benevolence

A Mann–Whitney $U$ test was conducted to determine if there was a significant difference in the benevolence of attitudes expressed towards individuals with mental illness between 2020 and 2023. The test revealed a significant difference in scores between the two samples, $U = 1{,}038{,}000$, $Z = 43.63$, $p = <.001$. The effect size ($r$) was .96, this is a large effect size following Cohen's guideline (1988). The 2020 sample reflected a median score of 26 (95% CI [25.892, 26.412]) and the 2023 sample reflected a median score of 13 (95% CI [12.878, 13.423]). These results suggest that attitudes expressed towards individuals with mental illness, which are benevolent have reduced significantly between 2020 and 2023.

#### Social restrictiveness

A Mann–Whitney $U$ test was conducted to determine if there was a significant difference in the social restrictiveness of attitudes expressed towards individuals with mental illness between 2020 and 2023. The test revealed there was not a significant difference in scores between the two samples, $U = 527{,}408$, $Z = −0.09$, $p = 0.925$. The effect size ($r$) was .002, this is a negligible effect size following Cohen's guideline (1988) and is congruent with the statistical significance. The 2020 sample showed a median score of 20 (95% CI [20.057, 20.615]) and the 2023 sample showed a median score of 20 (95% CI [20.034, 20.623]). These results suggest that attitudes expressed towards individuals with mental illness, which are socially restrictive have not changed between 2020 and 2023.

#### Community

A Mann–Whitney $U$ test was conducted to determine whether there was a significant difference in the Community Mental Health Ideology expressed towards individuals with mental illness between 2020 and 2023. The test revealed a significant difference in scores between the two samples, $U = 582{,}663$, $Z = 4.63$, $p = <.001$. The effect size ($r$) was .10, this is a small effect size following Cohen's guideline (1988). The 2020 sample reflected a median score of 20 (95% CI [19.364, 19.951]) and the 2023 sample reflected a median score of 19 (95% CI [18.508, 19.159]). These results suggest that the Community Mental Health Ideology expressed

towards individuals with mental illness has reduced between 2020 and 2023.

## Discussion

The objectives of this study were to evaluate whether stigmatising attitudes, behaviours, perceptions and knowledge of mental health in Ukraine have changed since the first time they were recorded. The study reflected conflicting results, with some measures showing desirable improvements in public attitudes and others reflecting negative changes over time.

First, our study found a decrease in benevolent attitudes towards individuals with mental health disorders along with a decrease in positive community mental health ideology. In congruence with these findings, the measure found that authoritarian attitudes towards individuals with mental health disorders have increased since the previous study. The only measure that did not show any changes since the previous study is the measure of socially restrictive attitudes towards individuals with mental health disorders. Our findings align with evidence from studies looking at mental health in England prior to the Time to Change program and the See Me campaign, which was delivered in Scotland (Mehta et al., 2009; Evans-Lacko et al., 2014; Henderson et al., 2019). Data from prior to and during the campaigns suggested that external factors may have influenced an increase in stigma towards individuals with mental health issues, in particular during times of economic hardship such as the economic recession. These conclusions may explain the current findings, as Ukraine has experienced a war since the previous study was undertaken, this has created economic instability and hardships for many, along with other impacts such as displacement (Kang et al., 2023; Seleznova et al., 2023). In addition to this, the study in England found less improvement in mental health stigma in more populated areas of the UK, such as London, than others, and noted that this area also has a higher prevalence of mental health diagnoses (Henderson et al., 2019). Similar results were found in a study in Taiwan, where it was found that individuals living in areas with a high density of psychiatric rehabilitation services had a higher desire for social distancing towards patients with depression (Tsai et al., 2020). Potential explanations for this have been attributed to a perpetuation of negative stereotypes through unstructured public exposure to individuals who are visibly unwell and exposure to narratives in the media, which portray violence committed by individuals with diagnoses (Henderson et al., 2019; Tsai et al., 2020). These findings might explain the findings of the current study, which is based in Ukraine, a country that is currently facing an increase in mental health needs in the general population as well as a lack of mental health service provisions as a result of the war (Kang et al., 2023; Seleznova et al., 2023). Following the findings above, this disruptive environment in Ukraine may partially explain a decline in positive attitudes towards individuals with mental health issues. However, further investigation would be required to confirm and untangle this relationship, both globally and in Ukraine.

The finding that stigmatising attitudes have worsened in Ukraine is not surprising given the other findings of this study, which demonstrated that knowledge of mental health in the general population also seems to have worsened since the original study. This means that the general public now reports a poorer understanding of mental health disorders. The link between knowledge of mental health disorders and the attitudes, which individuals hold towards those with mental health disorders has been well documented (Evans-Lacko et al., 2010; Thornicroft et al., 2022). Indeed,

it is argued that mental health knowledge plays an important role in mediating attitudes and behaviours relating to mental health stigma (Evans-Lacko et al., 2010). However, it is unclear as to why public knowledge of mental health in Ukraine has worsened over recent years. One potential explanation can be attributed to media coverage surrounding mental health in Ukraine over the past few years. News media is known to be a key source of information related to psychological and mental health issues (Gu and Ding, 2023; Zhang and Firdaus, 2024). Negative and misleading news and other media reports have been found to deepen misunderstandings and reinforce prejudices and stigmas against individuals with mental health disorders (Zhang and Firdaus, 2024). Moreover, during periods of humanitarian crises, pandemics and health emergencies, countries have been known to experience a surplus of false or misleading information pertaining to health and healthcare, a phenomenon referred to as "infodemics" (Nascimento et al., 2022). This suggestion has implications for the current study, given that Ukraine has experienced the impacts of both the COVID-19 Pandemic and the war perpetuated by Russia since the initial study. Finally, it is possible that the factors outlined above, alongside the decrease in access to mental health care, which was brought on by the war, have worsened public attitudes towards help-seeking and their faith in mental health services (Ronaldson and Henderson 2024). However, it should be noted that there has not been specific research undertaken in Ukraine to confirm these links.

While the literature outlined above provides potential explanations for the current findings pertaining to a decline in knowledge and in positive attitudes towards individuals with mental health disorders in Ukraine, they do not explain the opposing findings of this paper, which suggest a slight improvement in past, present and future intended behaviours. The behaviour scale used in this study has been used over recent years to evaluate the impact of specific anti-stigma campaigns. The majority of these report similar findings that increased campaign activity has shown a significant impact when stigma is measured through attitudes but not for behaviour (Evans-Lacko et al., 2014). In some instances, a positive change in behaviour is shown in the short term but does not last when measured at a later date (Pingani et al., 2021). Despite these findings, research conducted in the UK following the Time to Change campaign reflected a stronger initial increase in reported and intended behaviour than in knowledge and attitudes (Evans-Lacko et al., 2013). This is congruent with the findings of the current study. One of the suggestions put forth to explain this effect is that the focus of the campaigns in question might have had a stronger focused on behaviour change and a stronger emphasis on the public's active involvement than previous campaigns. It is possible that the nature of the campaigns in Ukraine resulted in a similar effect. However, to date, no studies were found which demonstrated the relationship between other external factors and behaviour, as has been done for attitudes in the research summarised earlier. Furthermore, it is not clear why research on mental health anti-stigma campaigns reflects an increase in reported behaviour, which is not associated with exposure to the campaign. This makes it difficult to draw conclusions to explain the conflicting findings, implications and suggestions for future research are discussed later.

### *Practical implications*

Given that mental health stigma is an issue which reduces uptake of mental health services and worsens health outcomes, the findings of the current study are of particular note (Livingston and Boyd 2010; Schnyder et al., 2017; Dubreucq et al., 2020; Thornicroft et al., 2022). The findings of this study that stigmatising attitudes and

knowledge of mental health are becoming worse, present valuable information for policymakers in Ukraine. While a number of international and Ukraine-based organisations are developing their own programmes to tackle mental health stigma, these findings suggest that a larger evidence-based campaign may be required to tackle this growing need (Media Column, 2023; Mental Health for Ukraine, 2023; Center for Health and Development 'Family Circle,' 2024; The Presidential Office of Ukraine, 2024). However, given that the majority of the existing anti-stigma projects have only been delivered over recent years, it may be that change is taking place slowly, further research will be required to determine this.

### Strengths and limitations

The current study has several noteworthy strengths. First, the conditions in which the study was replicated were identical to the initial study. Meaning that the same research company gathered their data using the same panel and recruiting methods, they also used the same established measures, which were left unaltered in this study to reduce any risk of inconsistencies. However, the study also had some limitations, which mimic those of the previous study. First, the survey was limited to those aged 60 or younger and was delivered online. While these limitations do not affect the significance of the differences we found over time, changes to these would be beneficial to achieve a more representative sample of the Ukrainian population. Second, while all of the measures used have been validated in English, they have been translated into Ukrainian by English- and Ukrainian-speaking mental health professionals and Ukrainian-speaking experts. This means that the findings of both studies may have been affected by the language used in translation. Third, the measures used in this paper do not distinguish between types of mental disorders when evaluating attitudes, knowledge and behaviours, which the public has of and towards those with mental disorders. Indeed, previous research has shown that public stigma towards those with disorders such as depression and anxiety is impacted differently over time when compared to public views towards individuals with diagnoses of more severe mental illnesses such as Schizophrenia (Schomerus et al., 2022). Further research that observes public views in relation to different disorders may be beneficial to understanding changes, or lack of changes, over time in Ukraine. Finally, the Ukrainian environment has shifted rapidly as a result of mental health reforms, the COVID-19 pandemic and the war perpetuated by Russia. This means that while preliminary conclusions have been drawn in this paper, it is not possible to fully identify the cause-effect relationship of these findings. While the current research provides unique insight into the changes in mental health stigma, future researchers should keep these external influences in mind when undertaking research on mental health stigma in Ukraine.

### Conclusions

Mental health stigma has been identified as a barrier for individuals with mental health disorders accessing healthcare, as well as a determinant of poor health outcomes for those affected. This study provides evidence of a concerning increase in stigmatising attitudes towards individuals with mental health disorders and a decrease in knowledge of mental health in Ukraine. On the flip side, the study shows improvements in the intended behaviours, which individuals have towards those with mental health disorders. These findings suggest a clear need for policymakers to consider implementing

evidence-based anti-stigma interventions targeting mental health knowledge and attitudes. Finally, the findings suggest a need to monitor the effectiveness of existing interventions in Ukraine over the coming years.

**Open peer review.** To view the open peer review materials for this article, please visit http://doi.org/10.1017/gmh.2025.40.

**Acknowledgements.** The authors thank Heiko Königstein for his unwavering support as Project Leader of the Mental Health for Ukraine project. The authors also thank Eleanor Quirke for her insight and helpful contributions to the development of this piece. Finally, they thank Be-It Health and Social Impact Agency and InMind for their collaboration and support in gathering the data used in this piece.

**Author contribution.** M.G., O.S. and V.K. planned and designed the study. M.G. and O.S. conducted the literature review. V.K. and M.G. analysed the data and interpreted the results. M.G. coordinated the write-up of the research and managed communication among authors. All authors reviewed and revised drafts of the manuscript prior to publication.

**Financial support.** The development of this publication has been made possible with the support of the Mental Health for Ukraine project. This is a project funded by the Swiss Agency for Development and Cooperation (SDC), with the aim of improving mental health outcomes within the Ukrainian population. The project is implemented by a consortium of organisations, which includes GFA Consulting Group GmbH, Implemental Worldwide C.I.C., the University of Psychiatry Zurich and the Ukrainian Catholic University Lviv.

**Competing interests.** This publication was developed by members of the Mental Health for Ukraine project team. The opinions and views presented in this publication represent those of the authors and may not necessarily represent the views of the Mental Health for Ukraine project team, the implementing consortia or the SDC.

**Ethical standards.** All participants in this survey gave their consent prior to participation. The procedures contributing to this work comply with the ethical standards of the relevant national committees and comply with the Helsinki Declaration of 1975, as revised in 2008.

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
