## [Reviewer Report]

Thank you for the opportunity to review this paper, which explores Ukrainians' attitudes towards and knowledge of mental health, as well as their intended behaviour in interactions with individuals with mental health issues. This paper contributes to the body of research on stigma in Eastern Europe, a region often overlooked, and provides first insights into how mental health stigma is modulated during crises like the ongoing war in Ukraine. Here are a few revisions that could enhance the paper:

1. The abstract mentions this as a longitudinal study, but the methods and data presented suggest an analysis of repeated cross-sectional nationwide surveys. I recommend adjusting the terminology throughout the paper to accurately reflect the research design and approach.

2. In the methods authors describe the recruitment process very briefly, it would be valuable to include more detail on the panel on which data was collected. For example, it is unclear how many respondents were in the panel for both runs of the 2020 and 2023 data collection? Or how many new respondents were recruited? How many reminders if any were sent to respondents to complete the survey and how were they processed?

3. The methods section describes the scales used, but additional information on the translation process from English to Ukrainian would be valuable. While the limitations section briefly mentions this, a more detailed explanation in the methods section would be beneficial.

4. A description of the analysis process along with a rationale for why such approach was chosen would also be a benefit in the methods section.

5. If possible, a table summarising the findings for all the section following sample demographics would allow for better readability of the findings.

---

## [Reviewer Report]

This is a paper whose importance lies in the setting of a middle income country, where such studies are relatively few; its use of standardised measures which have been widely used; and the use of timepoints which allow the study of how stigma is affected by population level stressors. Further work is needed throughout all sections of the paper, as detailed below.

Abstract

The first sentence is not in fact a sentence so needs rewriting. Add the figures here so that effect sizes of the changes over time are in the abstract. ‘Intended past, present and future behaviour’ should be rewritten as the RIBS does not cover intended past behaviour (indeed this would be hard to do).

Introduction

I suggest add that Ukraine is an upper middle income country- I see its status was upgraded for FY 2024 by the World Bank.

The introduction is focussed on the increase in common mental disorder and stigma reduction regarding CMD but there is nothing about severe mental illness which attracts more stigma, and noting about the impact of the war on services, the availability of which influences stigma. What was service provision like for each of CMD and SMI before the war and how have these been affected? Has SMI been covered at all by the stigma reduction efforts?

Methods

The subheading for the first section Design should be added. Cohorts are followed up and I do not think this was done, rather these are cross sectional surveys of 2 independent samples, correct? If so rewrite to clarify this.

Measures- information about the process of translation and psychometric testing in this population is needed.

I recommend use of and referral to the STROBE reporting guidelines for cross sectional studies.

Analysis: understandably there are big differences in the regional distribution of the two samples. Because of this and to identify demographic factors associated with the stigma outcome measures, linear regression modelling is needed including the variables shown in Table 1.

It is not clear whether the CAMI factors used are present in these data. The results are compared with those from the Time to Change evaluation, in which these factors were not identified, instead only two were identified (see Rüsch N, Evans-Lacko S, Henderson C Leese M, Flach C Thornicroft G. Public knowledge and attitudes as predictors of help-seeking and disclosure in mental illness. Psychiatric Services, 62:675-678, 2011). The authors may wish to consider a factor analysis using their data followed by regression analysis as suggested above.

Discussion

Regarding the worsened level of mental health related knowledge, it is worth considering the items in this scale in relation to change over time in access to services. If access has reduced over the two time points this may affect the item about knowing what to do about help seeking. Further, the current circumstances may affect therapeutic optimism and hence the level of agreement that medication and or psychological therapies can be helpful. For a discussion about this see Ronaldson A, Henderson C. Mental illness stigma in England: What happened after the Time to Change Programme to reduce stigma and discrimination. BPsych Open, (2024) 10, e199, 1–10. doi: 10.1192/bjo.2024.801.

The increase in reported contact is likely due to increased prevalence, and increased awareness of such contact through the efforts of Mental Health Ukraine. Increased intended contact was the first change seen during Time to Change, i.e. before later changes in the CAMI or MAKS (see Evans-Lacko S, Malcolm E, West K, Rose D, London J, Rüsch N, Little K, Henderson C. & Thornicroft G. Influence of Time to Change’s social marketing interventions on stigma in England 2009-11. British Journal of Psychiatry, 202 (suppl 55): s77-s88, 2013), and so a similar pattern may be occurring in Ukraine (so keep going!).

It is not the case that ‘no psychometrically tested instruments to assess behaviour which is stigmatising at population level had been developed until the Reported Intended Behaviour Scale in 2011 (Evans-Lacko et al. 2011).’ The RIBS is based on the social distance scale which has been very widely used with vignettes, for example see G Schomerus, C Schwahn, A Holzinger, P W Corrigan, H J Grabe, M G Carta, M C Angermeyer Evolution of public attitudes about mental illness: a systematic review and meta-analysis. Acta Psychiatr Scand. 2012 Jun;125(6):440-52. doi: 10.1111/j.1600-0447.2012.01826.x.

---

## [Reviewer Report]

Thank you for taking the time to revise your manuscript. The manuscript is now more clear and coherent and is ready for publication.